# Optimizing one-dose and two-dose cholera vaccine allocation in outbreak settings: A modeling study

**Tiffany Leung**[1], **Julia Eaton**[2], **Laura Matrajt**[1] *

**1** Vaccine and Infectious Disease Division, Fred Hutchinson Cancer Research Center, Seattle, Washington, United States of America, **2** School of Interdisciplinary Arts and Sciences, University of Washington, Tacoma, Washington, United States of America

* laurama@fredhutch.org

**Data Availability Statement:** All relevant data are within the manuscript and its Supporting information files.

**Funding:** TL and LM acknowledge support from the UK Foreign, Commonwealth and Development

## Abstract

### Background

A global stockpile of oral cholera vaccine (OCV) was established in 2013 for use in outbreak response and are licensed as two-dose regimens. Vaccine availability, however, remains limited. Previous studies have found that a single dose of OCV may provide substantial protection against cholera.

### Methods

Using a mathematical model with two age groups paired with optimization algorithms, we determine the optimal vaccination strategy with one and two doses of vaccine to minimize cumulative overall infections, symptomatic infections, and deaths. We explore counterfactual vaccination scenarios in three distinct settings: Maela, the largest refugee camp in Thailand, with high in- and out-migration; N'Djamena, Chad, a densely populated region; and Haiti, where departments are connected by rivers and roads.

### Results

Over the short term under limited vaccine supply, the optimal strategies for all objectives prioritize one dose to the older age group (over five years old), irrespective of setting and level of vaccination coverage. As more vaccine becomes available, it is optimal to administer a second dose for long-term protection. With enough vaccine to cover the whole population with one dose, the optimal strategies can avert up to 30% to 90% of deaths and 36% to 92% of symptomatic infections across the three settings over one year. The one-dose optimal strategies can avert 1.2 to 1.8 times as many cases and deaths compared to the standard two-dose strategy.

Office and Wellcome [grant number 215685/Z/19/Z]. The Scientific Computing Infrastructure at Fred Hutch was supported by the National Institutes of Health ORIP grant S10OD028685. The funders had no role in study design, data collection and analysis, decision to publish, or preparation of the manuscript.

**Competing interests:** The authors have declared that no competing interests exist.

## Conclusions

In an outbreak setting, speedy vaccination campaigns with a single dose of OCV is likely to avert more cases and deaths than a two-dose pro-rata campaign under a limited vaccine supply.

### Author summary

Cholera is a major global burden, with 21,000–143,000 deaths annually. The oral cholera vaccine (OCV) is given as two doses and is less effective in children under five years old than in those aged five years and older. However, supply is limited. In this study, we used mathematical models paired with optimization algorithms to find the optimal strategies to allocate vaccines in order to minimize the number of cases and deaths for three distinct settings in Chad, Thailand, and Haiti. We found that in the short term (1 year) when there is limited vaccine supply, it is optimal to vaccinate individuals over five years old with one dose and young children under five years old with two doses. Across the three settings, these optimal strategies prevent the most cases, save the most lives, and avert 1.2 to 1.8 times as many cases and deaths as the standard two-dose untargeted strategy. Our results support that if vaccine supply is limited, under an outbreak setting, mass vaccination campaigns with a single dose of OCV may prevent more cases and save more lives than would a standard two-dose untargeted campaign.

## Introduction

Cholera is an infectious disease caused by *Vibrio cholerae* with an estimated 1.3 to 4 million cases and 21,000 to 143,000 deaths annually [1], particularly in sub-Saharan Africa or displaced populations. Improved access to water, sanitation, and hygiene (WASH) is the mainstay of long-term cholera control and prevention. For the short to medium term, until WASH improvements are established, vaccination can serve as a key strategy in cholera control and prevention.

In 2013, the World Health Organization (WHO) created a global stockpile of oral cholera vaccine (OCV) to ensure access to OCVs in outbreak and humanitarian emergency situations [2]. The three main contexts under which countries may request doses from the global OCV stockpile include 1) emergency use to control an active outbreak, 2) emergency use to prevent outbreaks during a humanitarian crisis, and 3) preventive use for areas deemed "hot-spots" or where cholera is endemic [3]. OCV is administered in two doses, given two weeks apart, and is much less effective in young children than in adults. Vaccine efficacy for two doses is 42% in those under five years old and 72% in those five years and older against clinical cholera [4]. For one dose, vaccine efficacy is reduced to 8% for those under five years old and 57.5% for those five years and older [5, 6]. However, protection from both vaccine and natural cholera infection wanes over time [7–10]. Children under five years old bear the greatest burden of cholera [11].

The standard approach to mass vaccination campaigns of OCV is untargeted and administers two doses [12–15]. There remains, however, a global shortage of OCV. Countries have explored ways to maximize impact of limited vaccine supply. In April 2016 during the early stages of a cholera outbreak in Zambia, the Ministry of Health, in collaboration with Médicins sans Frontières (MSF) and the WHO, implemented a single-dose campaign as a way to stretch

vaccine supply [16, 17]. Eight months later in December, a second dose was administered when there was more stock [17]. A single-dose approach was also used for cholera outbreaks in South Sudan in 2015 and was effective at preventing medically attended cholera during an outbreak [18]. A one-dose OCV campaign would vaccinate more people in exchange for a lower level of protection.

The optimal use of vaccine depends on many factors, such as the objectives of a vaccination program, the number of doses available, vaccine efficacy and a region's local transmission dynamics. Local transmission dynamics may be influenced by region-specific patterns in migration, human mobility, and climate variability [19–23].

In this study, we use mathematical models paired with optimization algorithms to determine the optimal vaccine allocation strategies during an active outbreak. We minimized three metrics of infection and disease burden: cumulative infections, cumulative symptomatic infections, and cumulative deaths. We explore the impact OCV could have had in outbreaks in Chad, Thailand, and Haiti—three distinct settings representing an urban area, a refugee camp, and a region torn by natural disaster respectively. For these settings each with distinct local transmission dynamics, minimizing each metric produced a different optimal vaccination strategy. We quantify the impact of these optimal strategies through these metrics.

## Methods

Here we describe the methods we used for the study. We provide details of each outbreak setting. We describe the three mathematical models used per setting and highlight the distinctive features of each. These models were used to analyze how different settings can alter the optimal use of OCV following an outbreak. Next, we elaborate on the implementation of the vaccination campaign and the various vaccination strategies (including the optimal vaccine allocation strategy) we considered. We then describe the details of the optimization algorithm used to find the optimal vaccination strategy. Finally, we give details on the uncertainty analysis of our results.

### Mathematical models of cholera transmission for different outbreak settings

**Urban area: N'Djamena, Chad.**   Cholera has been present in Chad for decades [24]. In response to a large 2011 cholera epidemic in Chad, MSF worked closely with the Chadian Ministry of Health to try to limit the spread of the disease [25]. As part of that response, the MSF team and other health agencies collected case data in N'Djamena, the capital and largest city of Chad with a population of 993,500 in 2011, 19% of that are under five years old [25, 26]. It is unknown what triggered the epidemic, which resulted in 17,200 cases and 450 deaths, but it may be correlated with the occurrence of the rainy season [27, 28].

We constructed a transmission model with vaccination and two age groups: those below five years old, and those who are at least five years old. Susceptible (S) individuals are infected directly through contact with an infectious individual or indirectly through contact with contaminated water. They then become exposed but not yet infectious (E). After the latent period, they become infectious either with symptoms (I) or without symptoms (A). Eventually they recover (R) and are temporarily immune to infection until they become susceptible again. This model follows the general SEIRS framework, with two infectious compartments and a rainfall-modulated water reservoir (W) that tracks the concentration of *V. cholerae* (Fig 1A). We assume that asymptomatic infections are less infectious and shed less bacteria into the environment than symptomatic ones. Individuals may be unvaccinated or vaccinated with either one or two doses.

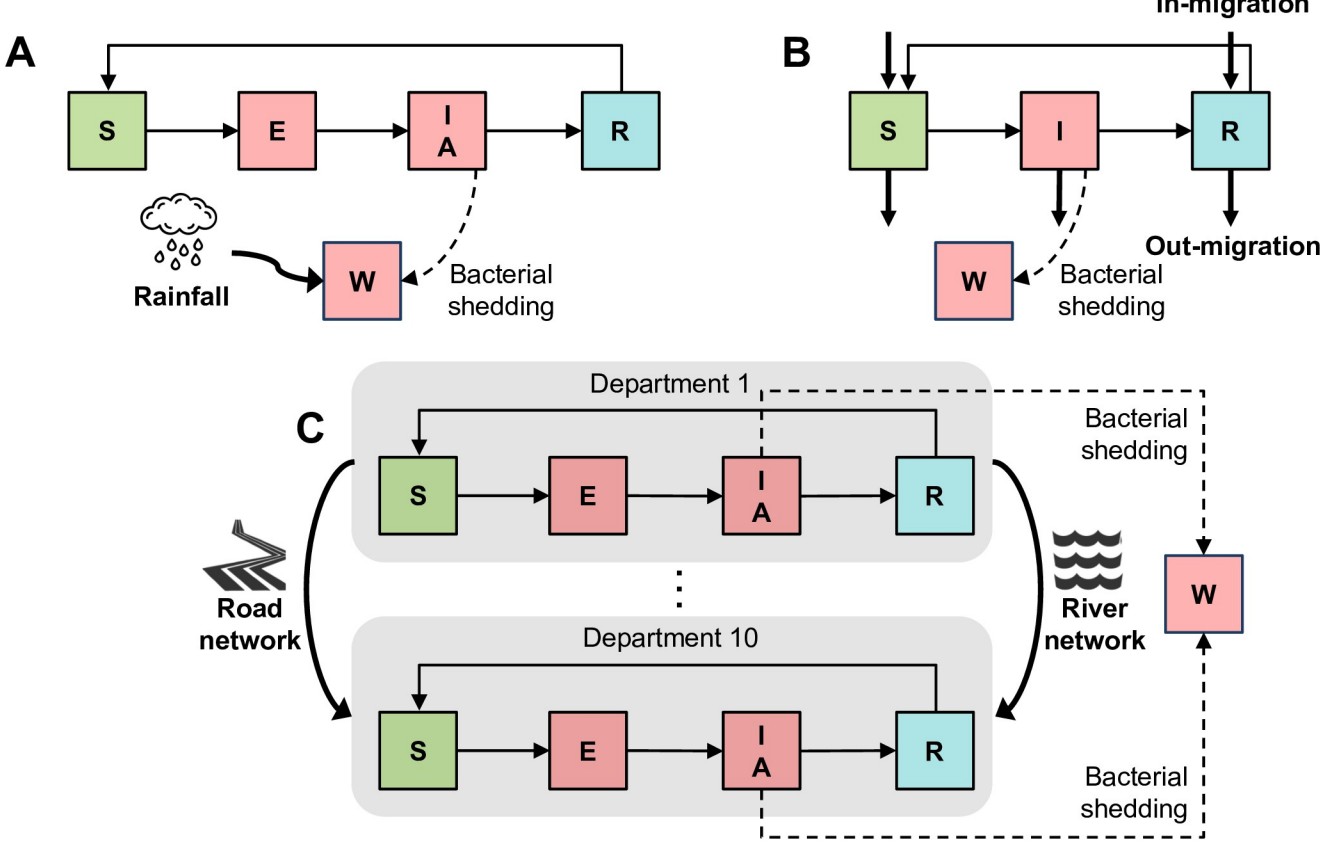

**Fig 1. General framework of the models.** The three models share common epidemiological states: susceptible (S), exposed but not yet infectious (E), infectious with symptoms (I), infectious without symptoms (A), and recovered (R). Infectious individuals shed bacteria into the water (W) (dashed line). Unique features of each model are shown in thick arrows. A: N'Djamena, Chad, a densely populated city. Transmission is influenced by rainfall. B: Maela, a refugee camp. Individuals may enter and exit the population through in- and out-migration. C: Haiti, where transmission is influenced by a road and water mobility network. These networks connect the ten departments. The age-structured and vaccinated analogues of these epidemiological states are omitted for clarity. See S1 Appendix for details.

Using incidence data collected by MSF, we calibrated this model without age structure or vaccination to the 2011 epidemic in N'Djamena, Chad [29]. All parameters were fixed except for the direct transmission coefficient, environmental transmission coefficient, reporting ratio, and the number of recovered individuals at the start of the simulation. Details on the model calibration are found in S1 Appendix.

**Maela refugee camp, Thailand.** Between 2005 to 2012, surveillance had identified four cholera outbreaks in Maela, the largest refugee camp in Thailand with high population density and migration [12, 30]. The second largest of these outbreaks occurred in 2010 with 362 confirmed cases [31]. At this time, the camp was home to approximately 45,000 refugees, and 13% were children under five years old [12, 31].

We adapted a previously developed deterministic model of cholera transmission with two age groups that was calibrated to the 2010 epidemic in Maela to compare the impact of various vaccination campaigns [31]. In the original version of this model, individuals are either susceptible, infectious, or recovered (Fig 1B). They may be unvaccinated, once vaccinated, or twice vaccinated. We calibrated the two age groups (below five years old, and five years and older) to Maela demographics [12]. This model differs from the other two because individuals may enter and exit the population through births, in- and out-migration, and natural death.

**A region torn by natural disaster: Haiti.**    Haiti comprises ten administrative departments, totaling 11 million inhabitants in 2010, with less than 12% under five years old [32]. In 2010 following a catastrophic earthquake, Haiti experienced its first recorded cholera outbreak in at least a century [33]. Poor access to water and sanitation in Haiti created a favorable condition for cholera to spread. Over the first two years of the epidemic, the Haitian Ministry of Public Health and Population reported more than 600,000 cases and over 7,000 deaths [34]. Since then, cholera has become endemic in Haiti [1].

We used a previously developed meta-population model of cholera transmission in Haiti. Each node in the model represents one of the ten administrative departments in Haiti [32]. The departments are connected through a river network and a road network, and within each department, individuals transition through a SEIR framework (Fig 1C). The road network was parameterized using a gravitational model and the river network was calibrated to Haiti's hydrology (full details given in [32]). Like in the model for Chad, infections may occur with or without symptoms. Environmental transmission is modulated by a sinusoidal function that represents the dry and rainy season. The mobility network (both rivers and roads) distinguishes this model from the models for Chad and Maela.

## Vaccination campaign

For each of the three models, we built and implemented a vaccination campaign that considered two age groups, those under five years old and those at least five years old. We considered a "leaky" vaccine that reduces the probability of infection upon exposure (reducing susceptibility) [35]. The vaccine effectiveness varied by age and by the number of doses received (S1 Table). Vaccine-induced immunity lasts an average of 4 years for two doses [7, 8], and 2 years for one dose [8]. As the duration of natural immunity lasts at least three years [9, 10], we assumed it to be equal to the duration of vaccine-induced immunity after two doses.

We explored counterfactual vaccination scenarios following cholera epidemics under varying levels of vaccination coverage, with enough vaccine to cover 10 to 100% of the population with a single dose (5 to 50% of the population with two doses). For each setting, we considered vaccination strategies that may allocate zero, one, or two doses to the different age groups (ages 0 to 4 years, and ages 5 years and older). The vaccination strategies considered include:

1. no vaccination (baseline);

2. two doses to individuals in both age groups allocated proportionally to their population (two-dose pro-rata, current standard);

3. one dose to individuals in both age groups allocated proportionally to their population (one-dose pro-rata);

4. two doses to children under five years old, and remaining doses allocated as one dose to individuals in the older group (mixed);

5. one dose to individuals five years and older (one-dose-over-five);

6. optimal allocation identified by our optimization routine (optimal).

We included the two-dose pro-rata strategy because this is the current standard, untargeted approach for mass OCV deployment worldwide [12–15]. We evaluated each strategy over one and three years. For each time horizon, we measured the performance of each vaccination strategy on three different metrics of disease burden: cumulative infections, cumulative symptomatic infections, and cumulative deaths.

Vaccine doses were distributed in mass vaccination campaigns which took place over one week per round, with 14 days between doses [3]. We assumed that protection began immediately after a dose is received, so that those who are allocated two doses were protected by the first dose until they received their second dose. Vaccination campaigns in Chad and Maela began two weeks after the simulation started when 50 and 20 cumulative cases were diagnosed respectively, while in Haiti, vaccination began at the start of simulation with 7300 cumulative cases, consistent with the available surveillance incidence data [32].

## Optimization

We applied optimization algorithms on the outcomes of our dynamic models of cholera transmission and vaccination for each setting to determine which vaccination strategy minimized disease metrics for each setting. We used a previously developed optimization routine [36, 37] that works in two steps: first, it performs an exhaustive grid search that allows us to explore the entire variable space within a 5% margin of vaccine allocation, followed by a heuristic global algorithm that uses the results from the grid search as initial conditions (description of the optimization routine can be found in S1 Appendix).

## Uncertainty analysis

Once a solution was deemed optimal, we compared its performance to other strategies (the two-dose pro-rata, the one-dose pro-rata, mixed, and the one-dose-over-five). We examined the uncertainty in the metrics of disease burden (total infections, symptomatic infections, and deaths) arising from the uncertainty around the values of key model parameters. For these parameters, we sampled 1000 parameter sets from pre-determined distributions and recalculated the reductions in the metrics of disease burden. We report here the 95% uncertainty intervals (full details, including the parameters that were varied, can be found in S1 Appendix).

## Results

### Optimal vaccine allocation by metrics of disease burden

In the next sections, we provide results for a time horizon of one year. The optimal strategies that minimized cumulative infections, symptomatic infections, and deaths over one year were similar across Chad, Maela, and Haiti. When minimizing total and symptomatic infections, the optimal allocation strategies were largely identical in all three settings: Prioritize those at least five years old with a single dose of vaccine (Fig 2A, 2B, 2D, 2E, 2G and 2H). Once everyone in this group has received one dose, these optimal strategies allocated any remaining vaccine (above 80% vaccination coverage) differently by setting: i) in Chad, to those under five years old with two doses (Fig 2A and 2B); ii) in Haiti, as a second dose to those at least five years old (Fig 2G and 2H); and iii) in Maela, either of these strategies depending on whether total or symptomatic infections were minimized (Fig 2D and 2E). When minimizing deaths, the optimal strategies for Chad and Maela were the same as the ones which minimized symptomatic infections (Fig 2C and 2F). In contrast, in Haiti, the optimal strategy to minimize deaths prioritized the age groups in reverse order: Allocate two doses to children under five years old before distributing one dose to those at least five years old (Fig 2I).

As expected, reactive vaccination in an outbreak setting can substantially curtail the epidemic. Without vaccination the prevalence of cases peaked at 23, 60, and 179 cases per 100,000 for Chad, Maela, and Haiti respectively (Fig 3). Using the optimal strategies that minimized symptomatic infections (Fig 2B, 2E and 2H), vaccination lowered the peak prevalence the

          

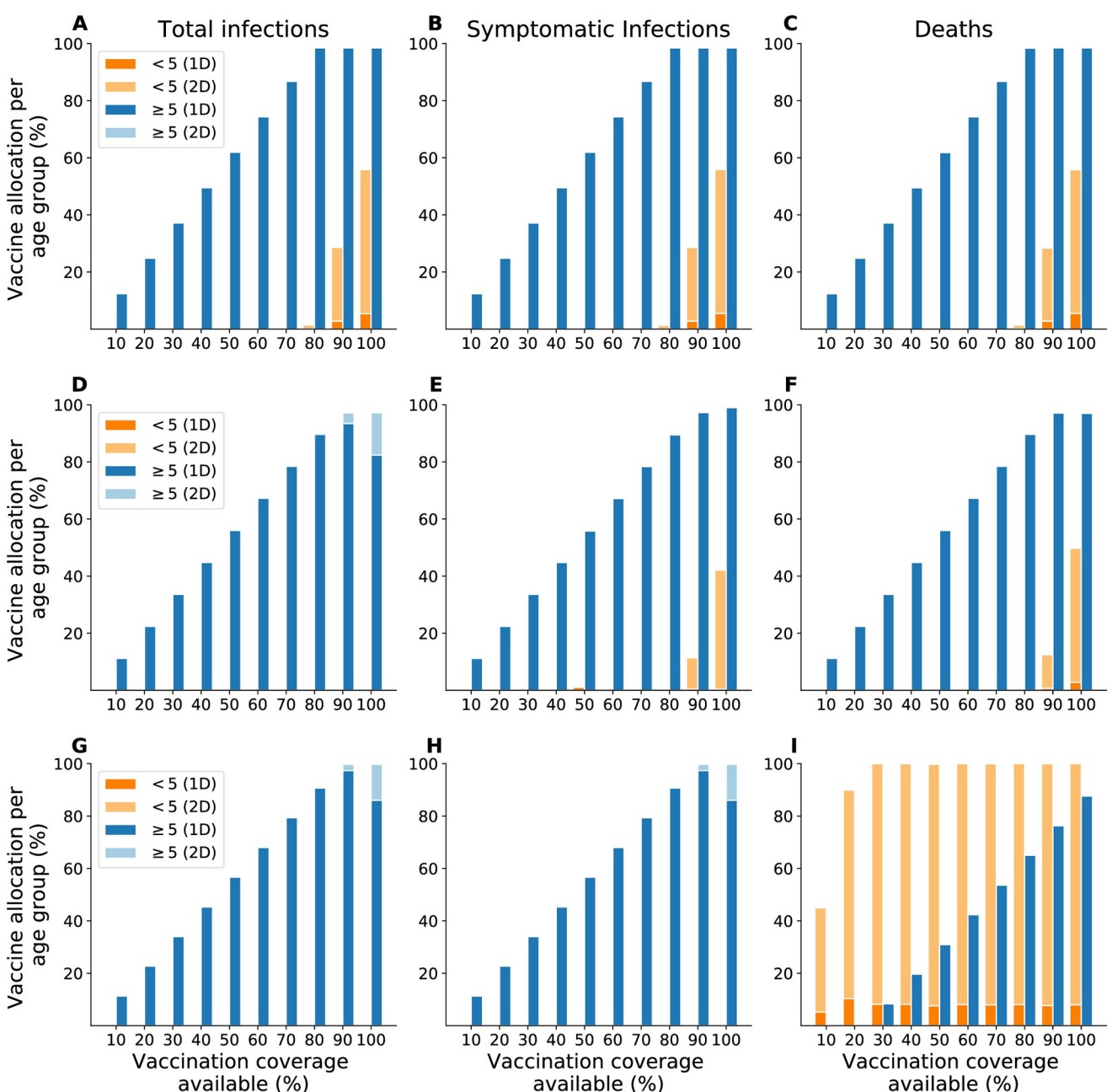

**Fig 2. Optimal vaccine allocation strategies to minimize total infections (left), symptomatic infections (center), or deaths (right) over one year.** A–C: Chad. D–F: Maela. G–I: Haiti. We considered enough vaccine to cover 10–100% of the population with a single dose.

most in Maela (Fig 3B) and least in Haiti (Fig 3C). At 50% coverage, these optimal strategies lowered the peak prevalence (per 100,000) to 10 in Chad, 12 in Maela, and 133 in Haiti (reduction by 50%, 80%, and 26% respectively) (Fig 3). Next, we evaluated how the optimal strategies for each objective performed against other vaccination strategies as measured by the percentage averted for the three metrics of disease burden: total infections, symptomatic infections, and deaths.

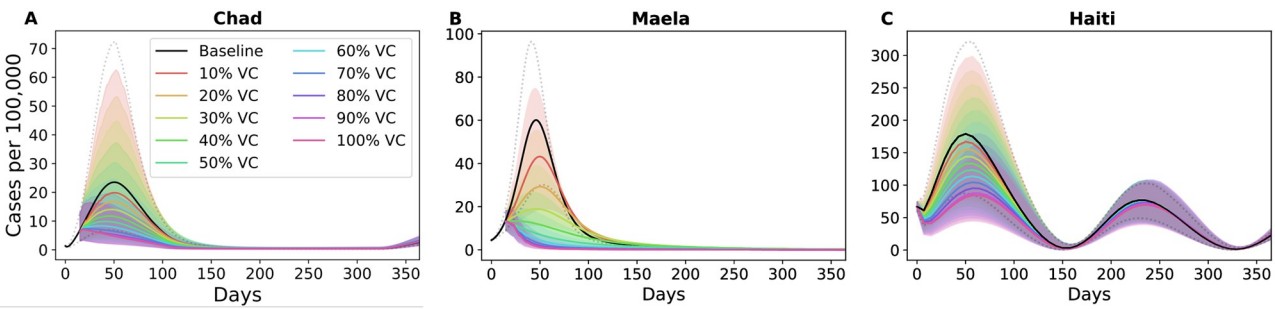

**Fig 3. Prevalence of cases per 100,000 over one year.** Prevalence of cases using an optimal vaccine allocation that minimized symptomatic infections over different vaccination coverage for A: Chad, B: Maela, and C: Haiti.

## Performance of the optimal strategies in Chad, Maela, and Haiti

In Chad, the optimal strategies that minimized each metric of disease burden encompassed the one-dose-over-five strategy. These optimal strategies could have reduced up to 68% of total and symptomatic infections and 61% of deaths compared to no vaccination (Fig 4A, 4B and 4C). The optimal strategies minimizing total infections, symptomatic infections, and deaths maximally outperformed the two-dose pro-rata strategy at 80% coverage: by 23%, 23%, and 17% respectively. Compared to the one-dose pro-rata strategy, these optimal strategies averted up to 8% (at 80% coverage) more total and symptomatic infections and averted up to 6% (at 80% coverage) more deaths (Fig 4A, 4B and 4C).

In Maela, the optimal strategies to minimize each metric also encompassed the one-dose-over-five strategy, outperforming all other vaccination strategies (Fig 4D, 4E and 4F). These optimal strategies had larger reductions than the two-dose pro-rata strategy by up to 30% (and bigger than the one-dose pro-rata strategy by up to 8%). At 60% vaccination coverage, these optimal strategies had averted at least 75% of total infections, symptomatic infections, and deaths compared to no vaccination.

In Haiti, the optimal strategies reduced at most 36% of total and symptomatic infections and 31% of deaths (Fig 4G, 4H and 4I). The optimal strategies outperformed the two-dose pro-rata strategy by a small margin (up to 9% of total and symptomatic infections and 8% of deaths) and the one-dose pro-rata strategy by a smaller margin still (up to 3% of total and symptomatic infections and 6% deaths).

Our analysis suggests that optimal vaccination strategies will be most effective in Maela (with the highest percentage of total infections, symptomatic infections, and deaths averted), followed by Chad and Haiti (Fig 4). For example, at 80% vaccination coverage in Maela, Chad, and Haiti, the optimal strategies that minimized deaths averted 87%, 55%, and 25% of deaths, respectively (Fig 4C, 4F and 4I).

## Comparison of the vaccination strategies across settings

The vaccination strategies that allocated a single dose of OCV outperformed the ones that allocated two doses for all three metrics and settings (Fig 4A, 4B, 4C, 4D, 4E, 4F, 4G and 4H), except for averted deaths in Haiti (Fig 4I). This suggests that the timing of the vaccination campaign plays a role in the optimal use of vaccine. When vaccination is started late, when many infections have already occurred (like in our scenario for Haiti) direct protection of those at highest risk with the most efficacious vaccine (two-dose strategies) is preferred. In contrast, when vaccination started early, vaccinating as many individuals as possible (single-dose

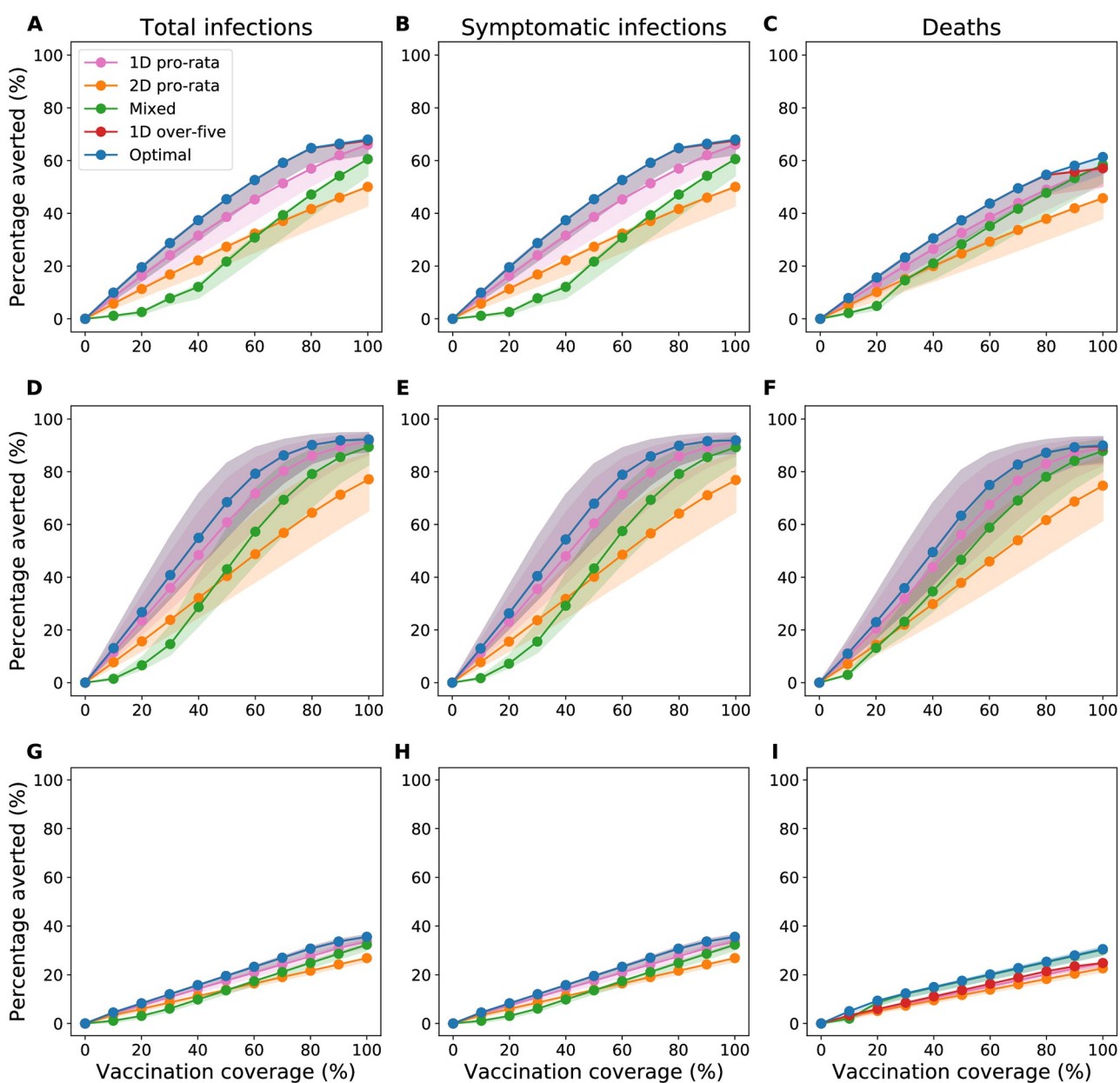

**Fig 4. Reductions in the metrics of disease burden from the vaccination campaigns over one year.** Percentage of total infections (left), symptomatic infections (center), and deaths (right) averted over one year for A–C: Chad, D–F: Maela, and G–I: Haiti. We considered enough vaccine to cover 10–100% of the population with a single dose.

strategies) to curtail transmission is better. Aside from the optimal strategy, the one-dose pro-rata strategy had the biggest impact with respect to all metrics (Fig 4A, 4B, 4C, 4D, 4E, 4F, 4G and 4H). The two-dose pro-rata strategy outperformed the mixed strategy at lower vaccination coverage, while the latter outperformed the former at higher coverage. This crossover occurred at between 30% to 60% vaccination coverage (Fig 4).

Comparing the pro-rata vaccination strategies, one-dose outperformed a two-dose allocation, irrespective of setting, metric of disease burden, and vaccination coverage. For example, for averted deaths, the one-dose pro-rata strategy outperformed the two-dose pro-rata strategy

by up to 22%, 12%, and 2% of deaths in Maela, Chad, and Haiti respectively (Fig 4C, 4F and 4I). While this difference was large in settings where vaccination started early (Maela) it became insignificant in Haiti, where vaccination was assumed to start very late with respect to the epidemic curve, demonstrating that early vaccination is key to maximizing impact on reducing disease burden. Furthermore, in Maela and Chad, where vaccination started early in the epidemic when relatively fewer cases were diagnosed, vaccination prevented up to 90% and 61% of deaths respectively, compared to 28% in Haiti, where vaccination began later.

## Short-term versus long-term impact

We investigated how the optimal strategies for each setting may change over a longer time horizon of 3 years. Recurring epidemics corresponded to seasonally forced transmission in both models for Chad and Haiti. In Maela, one case was re-seeded 1.5 years (chosen as halfway through the time horizon) after the first identified case.

**Optimal strategies over a three year time horizon.** When minimizing total and symptomatic infections, the optimal strategies in Maela, Chad, and Haiti all prioritized individuals at least five years old (Fig 5A, 5B, 5D, 5E, 5G and 5H). In Maela and Chad, this age group was allocated a single dose when vaccine supply was low (below 40% vaccination coverage) and a mix of one and two doses with higher vaccine supply (Fig 5A, 5B, 5D and 5E). In Haiti, the optimal strategies allocated two doses to those at least five years old regardless of vaccine availability (Fig 5G and 5H).

When the objective was to minimize deaths over three years, the optimal strategies varied across Chad, Maela, and Haiti. In Chad, the optimal strategy was similar to the mixed strategy, prioritizing children under five years old with mostly two doses for all vaccination coverage. Once these children were vaccinated (above 30% vaccination coverage), the optimal strategy allocated the remaining vaccine to those at least five years old with a mix of one and two doses (Fig 5C). In Haiti, the optimal strategy first targeted children under five years old with mostly two doses before allocating two doses to those at least five years old with any remaining vaccine (Fig 5I). In Maela, the optimal strategy prioritized the age group at least five years old with one dose (below 60% vaccination coverage) and with two doses (60% coverage and above), and allocated some vaccine to children under five years old (40% to 70% coverage) (Fig 5F).

**Performance of vaccination strategies over three years.** While the absolute impact was larger over three years than one year, the impact of the vaccination strategies relative to no vaccination, measured by percentage of infections and deaths averted, was smaller at three years than at one year (Fig 6). In Maela, the re-introduced case did not produce any additional outbreaks, whereas there were multiple epidemic waves in Chad and Haiti. Across the settings, vaccination strategies over the longer term had the biggest impact in Maela, followed by Chad and Haiti (same order as over the shorter term).

The performance differences between vaccination strategies over three years were smaller compared to those over one year. When minimizing transmission (total and symptomatic infections) over three years, the optimal strategies maximally outperformed the other strategies (at 100% vaccination coverage) by 8% in Chad (Fig 6A and 6B), 14% in Maela (Fig 6D and 6E), and 2% in Haiti (Fig 6G and 6H). When minimizing deaths, the optimal strategy at 80% vaccination coverage for Chad, Maela, and Haiti averted 25%, 56%, and 10% of deaths respectively (Fig 6C, 6F and 6I). At this vaccination coverage, the differences in deaths averted between the highest (optimal) and lowest (one-dose pro-rata) performing strategies were 8% in Chad, 13% in Maela, and 3% in Haiti. Over the longer term, the one-dose pro-rata strategy did not always outperform the two-dose pro-rata strategy. This is expected, as protection from

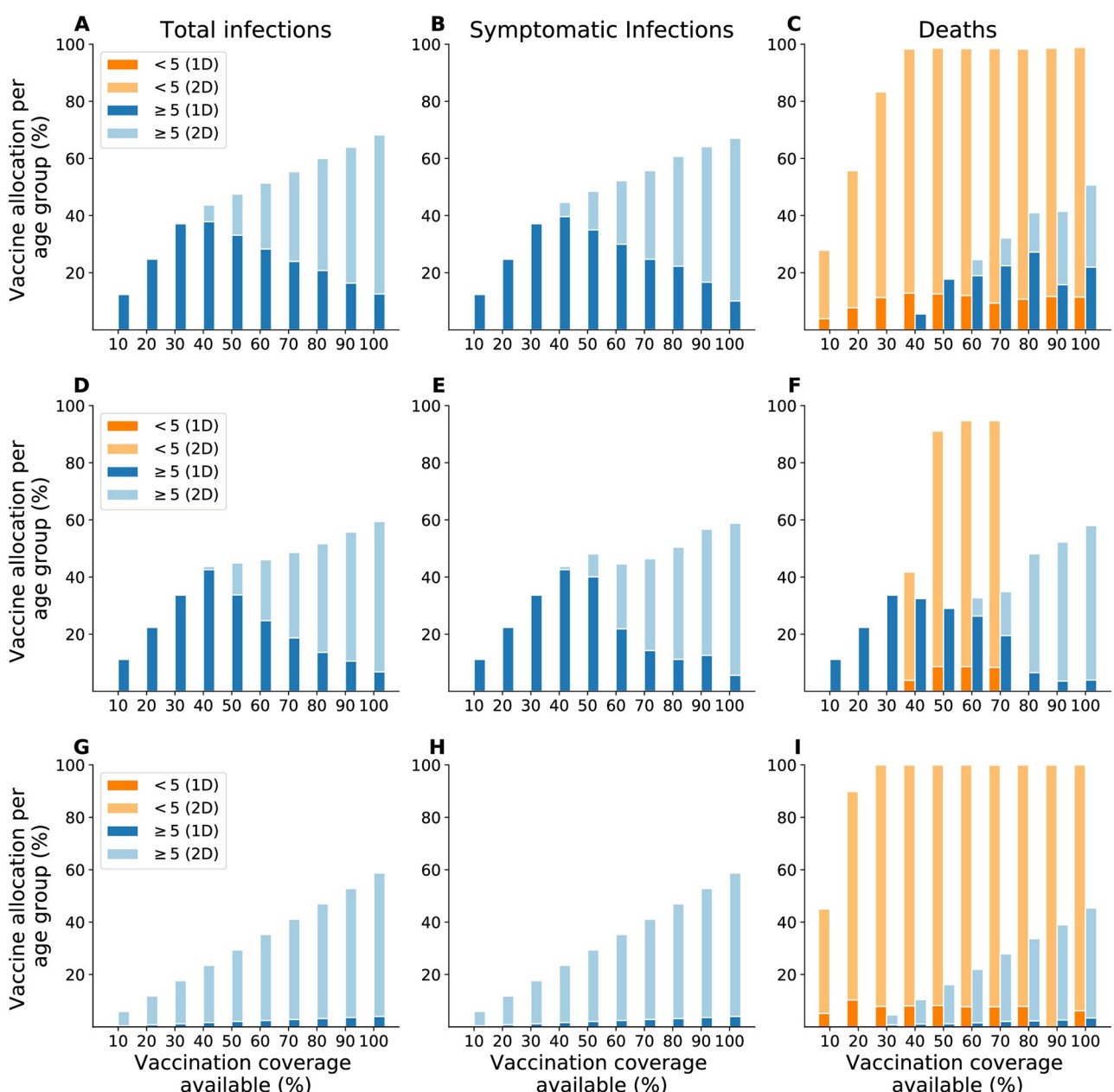

**Fig 5. Optimal vaccine allocation strategies to minimize total infections (left), symptomatic infections (center), and deaths (right) over three years.** A–C: Chad. D–F: Maela. G–I: Haiti. We considered enough vaccine to cover 10–100% of the population with a single dose.

a single dose was assumed to wane after two years (compared to four years of protection from two doses).

The optimal strategies over one and three years indicate that it is best to allocate one dose under limited vaccine supply in an outbreak setting for the highest reductions in disease and deaths over the short-term. However, when more vaccine is available later, a second dose can be allocated to prevent further outbreaks.

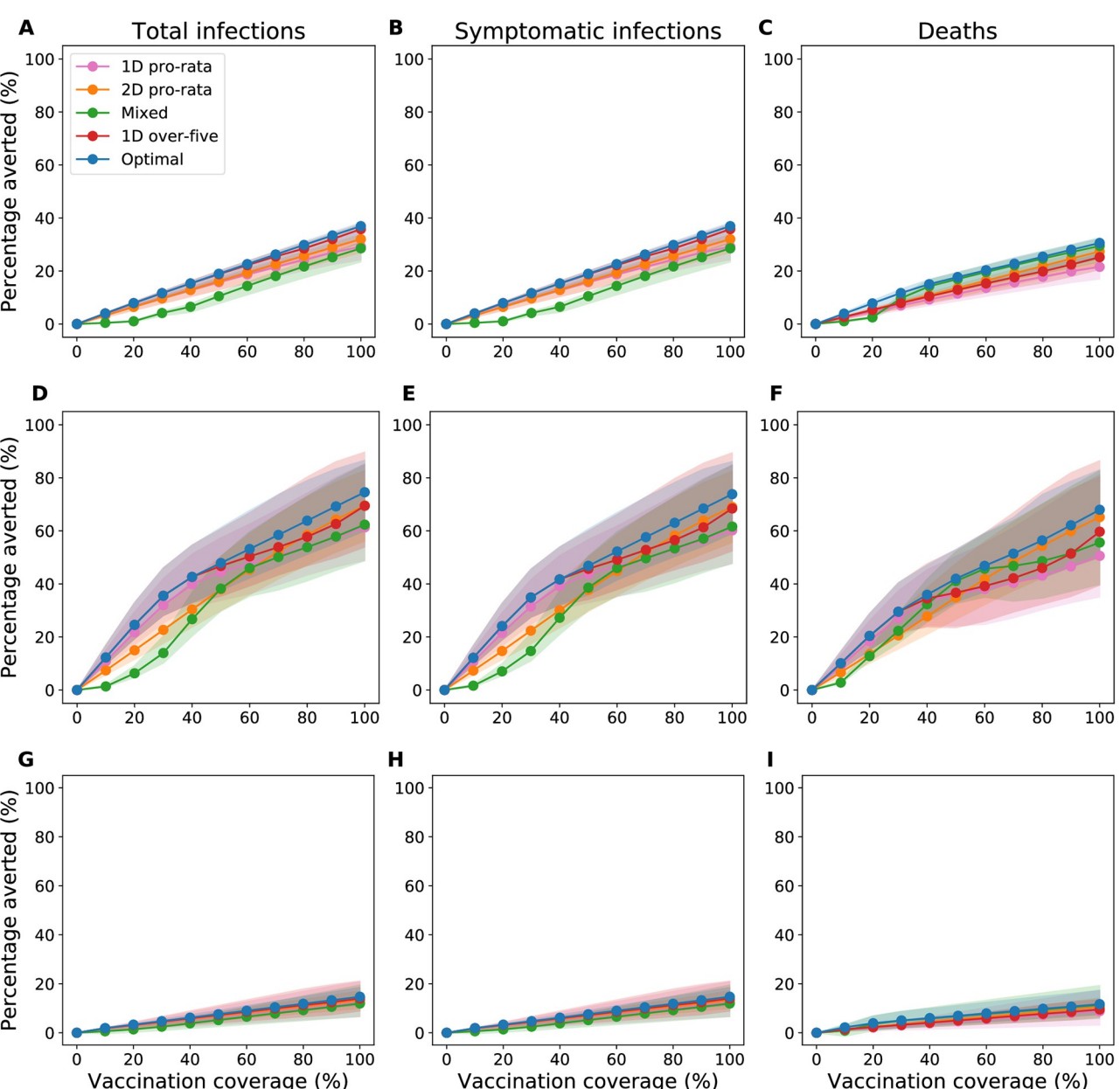

**Fig 6. Reductions in metrics of disease burden from the vaccination campaigns over three years.** Percentage of total infections (left), symptomatic infections (center), and deaths (right) averted over three years for A–C: Chad, D–F: Maela, and G–I: Haiti. We considered enough vaccine to cover 10–100% of the population with a single dose.

## A vaccine that reduces the likelihood of symptoms upon infection

In this section, we investigated how the optimal strategies may change if a vaccine reduced the likelihood of having symptoms upon infection (disease-reducing) instead of one that reduced susceptibility to infection. We illustrate the results with the model calibrated for Chad.

**Optimal vaccination strategies.** Over one year, the optimal strategies that minimized total infections, symptomatic infections, and deaths were the same for both vaccines (S1 Fig), prioritizing a single dose to those at least five years old before allocating two doses to the

younger age group. Over three years, the optimal strategies for each objective using a disease-reducing vaccine were slightly different to those using a susceptibility-reducing vaccine. When minimizing transmission (total or symptomatic infections), the optimal strategies for both vaccines prioritized a single dose to those at least five years old. For the disease-reducing vaccine, a second dose was allocated to the same group when coverage exceeded 70% (S2(A) and S2(B) Fig), whereas for the susceptibility-reducing vaccine, it was allocated sooner (above 30% coverage) (Fig 5A and 5B). When minimizing deaths, the optimal strategies prioritized two doses to children under five years old for both vaccines (S2(C) Fig and Fig 2C). Any remaining vaccines were allocated to those at least five years old as a single dose (with the disease-reducing vaccine) or as a mix of one and two doses (susceptibility-reducing vaccine).

**Comparison of optimal strategies using two different vaccines.**   Here we compare the performance of the optimal strategies using the two different vaccines. Not surprisingly, the susceptibility-reducing vaccine outperformed the disease-reducing vaccine when minimizing total infections by up to 23% and 19% over one and three years respectively (Fig 4A, S3(A) Fig, Fig 6A, and S4(A) Fig). When minimizing symptomatic infections and deaths, however, the optimal strategies using a disease-reducing vaccine outperformed ones using a susceptibility-reducing vaccine by up to 2% over one year and 12% over three years (Fig 4B and 4C, S3(B) and S3(C) Fig, Fig 6B and 6C, S4(B) and S4(C) Fig).

**Comparison of optimal strategies to other vaccination strategies.**   Here we compare the optimal strategies using the disease-reducing vaccine with other vaccination strategies. There was no qualitative difference in the performance ranking among the various vaccination strategies over one year between the two vaccines. The one-dose strategies still broadly outperformed two-dose strategies for all metrics of disease burden over one year and for averting total and symptomatic infections over three years (S3 and S4 Figs). The differences in performance between the strategies with the highest and lowest reductions for total infections, symptomatic infections, and deaths were up to 17%, 24%, and 17% over one year respectively (or 4%, 11%, and 10% over three years respectively) (S3 and S4 Figs).

## Discussion

We paired mathematical models of cholera transmission and vaccination with optimization algorithms to determine the optimal vaccination strategies and their impact on various metrics of disease burden. We explored vaccination campaigns that included the use of one dose of OCV, which is half of the recommended two-dose regimen. We considered two target populations by age (under five years old, and five years and older), different levels of vaccination coverage, three distinct cholera outbreak settings, and a time horizon of one and three years. We compared these optimal strategies to other vaccination strategies.

We found that, over the short term, strategies allocating one dose of vaccine to those five years and older reduced the most cumulative infections, symptomatic infections, and deaths for all three outbreak settings in Chad, Maela, and Haiti. In Haiti, the optimal allocation strategies also fully protected children under five years old with two doses to minimize deaths. Our results suggest that over a shorter term (one year), strategies with a one-dose allocation to those aged five years and older, for whom one dose of vaccine is more effective, outperform those allocating two doses irrespective of setting, metric of disease burden, and vaccination coverage. Further, our results show that if reactive vaccination campaigns start late with respect to the epidemic curve, when there are already a significant number of cases (Haiti in our settings) then it is still optimal to give one dose to those aged five years and older, but to also give full protection to the youngest children. This would stretch the vaccine supply

considerably, and of course, once vaccine supply is replenished, vaccination campaigns with a second dose for long-term protection are desired.

A modeling study has found that a one-dose vaccination campaign would avert 1.1 to 1.2 times as many cases and deaths as a two-dose campaign in an outbreak setting over the short term [38]. This is consistent with our finding that optimal strategies for each objective, which allocated a mix of one and two doses, would avert up to 1.2 to 1.8 times as many cases and deaths as a two-dose pro-rata strategy over the course of one year. Similarly, the one-dose pro-rata strategy would avert 1.08 to 1.5 times as many cases and deaths as the two-dose pro-rata strategy across the three outbreak settings. Different to their study, we used two age groups and age-specific case fatality rates. In particular, we assumed that the case fatality rates for children under five years old were about 5 times higher than individuals who were older [39]. Another modeling study analyzed the impact of vaccination campaign timing and one-dose vaccine effectiveness and found that timing had a bigger impact on case counts than one-dose effectiveness [31]. This is consistent with our finding that the impact of vaccination was smallest in Haiti, where vaccination began later during the exponential phase of the epidemic.

Our results of the optimal use of OCV are consistent with previous studies for other infectious diseases that have demonstrated that the optimal use of vaccine depends on coverage, vaccine efficacy and the timing of vaccination with respect to the epidemic curve, favoring direct protection of high-risk groups if vaccination starts too late [40–42]. However, the procurement and deployment of OCV usually include significant delays. There is a median delay of three months between the declaration of an emergency and the start of the first round of vaccination [3]. Delay times range widely—the time from the first laboratory confirmation of cholera (or occurrence of humanitarian emergency) to the receipt of the official OCV request spans between 12 and 206 days [3]. These delays in vaccine shipment may result in campaigns starting after the outbreak is over [43]. Further, our results are in line with previous modeling that explored the optimal use of vaccine when multiple doses are required for other pathogens, finding that fractional strategies are optimal when the protection afforded by a lower-than-recommended dose is sufficiently high [37, 44–46]. Besides cholera, other diarrheal diseases pose significant burden. In 2016, rotavirus infection caused more than 258 million episodes of diarrhea among children under five years old globally [47]. Rotavirus vaccines are also delivered in multiple doses, and some studies have suggested that an incomplete series of the rotavirus vaccine may provide sufficient protection against severe disease and hospitalizations [48–50]. It is important to consider that the optimal allocation of vaccine may change under different epidemiological contexts. In the present work we showed the optimal allocation of OCV under the context of emergency use to control an active outbreak (in epidemic situations). However, cholera outbreaks range widely depending on the region (short outbreaks are commonly seen in Africa while cholera is endemic in Bangladesh), and the optimal vaccine allocation may be different if used to prevent outbreaks or in endemic areas with hot spots.

The similar optimal vaccination strategies across three different settings and under varying degrees of coverage provide reassurance that these results are robust to a range of transmission dynamics. However, there are limitations to our work. As with all mathematical models, our models have simplifying assumptions on the true transmission dynamics and use parameter values that are subject to some uncertainty. In particular, we assumed homogeneity in mixing patterns and probability of a symptomatic case given infection. We calibrated the Chad model to incidence data. However, the burden of cholera is under-reported [51, 52]. Countries may suppress reports of cholera cases due to the perceived influence cholera has on tourism and export industries [53]. Additionally, the WHO case definition for cholera currently excludes children under five years old [54]. This exclusion can further obscure the true burden of cholera, especially in areas where children face increased cholera incidence. Importantly, the case

fatality ratio for cholera differs by age and geography. We assumed a fixed multiplier that increased childhood mortality. In reality, this multiplier is likely to change for different regions with different healthcare systems and access to clean water. Further studies are needed to truly evaluate the burden of cholera in young children. In particular, serological studies [55] might be useful to determine the burden of different diarrheal diseases in a particular region. We did not consider other possible confounders (e.g., differences in waning immunity and pre-existing immunity in endemic versus epidemic settings, the use of other interventions in tandem with vaccination) which could affect the population impact of vaccination.

There is some uncertainty around the vaccine efficacy of one dose of OCV. Trials of one-dose vaccine efficacy of OCV have demonstrated good protection for individuals over five years old, and low to no protection for children below five years old in Bangladesh where cholera is endemic [5, 6]. One-dose vaccine effectiveness from case-control; test-negative, case-control; and case-cohort studies ranges widely from 33% to 87%, and some of these estimates were not statistically significant [13, 18, 56, 57]. Understanding how that protection translates to different settings, such as in Haiti where cholera is newly endemic, should be a priority. We also made a simplifying assumption in our models that all children under five years old were eligible to be vaccinated. In reality, the vaccine is not approved for children under one year old.

Despite the counterfactual nature of our vaccination campaigns, our results can provide a basis for future cholera outbreaks, especially under settings with poor water infrastructure and constrained resources. We explored hypothetical vaccination campaigns that took place two weeks after the first case was identified (Chad and Maela) and during the exponential phase of the epidemic (Haiti). In reality, in emergency situations, the time from vaccine request to vaccine delivery are often long [3]. In emergency situations, the median time between the first laboratory confirmation (or occurrence of a humanitarian emergency) to one week after the first round of vaccination when immunity develops was 66 days [3]. Our results point to the need to reduce these lag times in order to increase the impact of vaccination campaigns. A one-dose vaccine allocation would reduce this logistical hurdle, and it would not preclude a second dose once there is more vaccine supply. However, in places where there is a lot of population migration, a long interval between the first and the second dose might result in having a proportion of the population receiving only one dose. In the event that the risk of future epidemics persist, recurrent vaccination campaigns may be required to maintain protection over time.

## Supporting information

**S1 Appendix. Appendix: The mathematical models.**
(PDF)

**S1 Fig. Optimal vaccine allocation strategies for different goals in Chad over one year using a vaccine that reduces the probability of symptoms upon infection.** A: total infections. B: symptomatic infections. C: deaths. We considered enough vaccine to cover 10–100% of the population with a single dose.
(TIF)

**S2 Fig. Optimal vaccine allocation strategies for different goals in Chad over three years using a vaccine that reduces the probability of symptoms upon infection.** A: total infections. B: symptomatic infections. C: deaths. We considered enough vaccine to cover 10–100% of the population with a single dose.
(TIF)

**S3 Fig. Reductions in metrics of disease burden from vaccination campaigns in Chad over one year with a vaccine that reduces the probability of symptoms upon infection.** We considered enough vaccine to cover 10–100% of the population with a single dose.
(TIF)

**S4 Fig. Reductions in metrics of disease burden from vaccination campaigns in Chad over three years with a vaccine that reduces the probability of symptoms upon infection.** We considered enough vaccine to cover 10–100% of the population with a single dose.
(TIF)

**S1 Table. Description of parameters related to the vaccine and disease metrics.**
(PDF)

## Acknowledgments

We thank Joshua Havumaki, Rafael Reza, Marisa Eisenberg, and Andrew Azman for providing access to research-related materials that were key to our analysis. We thank Dobromir Dimitrov for his helpful feedback on the manuscript.

## Author Contributions

**Conceptualization:** Tiffany Leung, Laura Matrajt.

**Data curation:** Tiffany Leung, Julia Eaton, Laura Matrajt.

**Formal analysis:** Tiffany Leung, Laura Matrajt.

**Funding acquisition:** Laura Matrajt.

**Investigation:** Tiffany Leung, Laura Matrajt.

**Methodology:** Tiffany Leung, Julia Eaton, Laura Matrajt.

**Project administration:** Laura Matrajt.

**Resources:** Laura Matrajt.

**Supervision:** Laura Matrajt.

**Validation:** Tiffany Leung, Laura Matrajt.

**Visualization:** Tiffany Leung.

**Writing – original draft:** Tiffany Leung.

**Writing – review & editing:** Tiffany Leung, Julia Eaton, Laura Matrajt.

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
