## [Decision Letter · Decision Letter 0]

15 Feb 2022

Dear Dr. Matrajt,

Thank you very much for submitting your manuscript "Optimizing one-dose and two-dose cholera vaccine allocation in outbreak settings: A modeling study" for consideration at PLOS Neglected Tropical Diseases. As with all papers reviewed by the journal, your manuscript was reviewed by members of the editorial board and by several independent reviewers. The reviewers appreciated the attention to an important topic. Based on the reviews, we are likely to accept this manuscript for publication, providing that you modify the manuscript according to the review recommendations. 

Your manuscript has been reviewed by experts in cholera vaccinology. Each of the reviewers has requested some minor revisions to help clarify the methodology and some additional discussion regarding the limitations of the analyses presented. Their comments are appended here (comments from Reviewer #1 are shown below and comments from Reviewer #2 are attached). Please give their feedback careful consideration in drafting a revised manuscript.

Sincerely,

James Michael Fleckenstein, M.D.

Associate Editor

Sharon Tennant

Deputy Editor

Your manuscript has been reviewed by experts in cholera vaccinology. Each of the reviewers has requested some minor revisions to help clarify the methodology and some additional discussion regarding the limitations of the analyses presented. Their comments are appended here. Please give their feedback careful consideration in drafting a revised manuscript.

Reviewer's Responses to Questions

**Key Review Criteria Required for Acceptance?**

**Methods**

-Are the objectives of the study clearly articulated with a clear testable hypothesis stated?

-Is the study design appropriate to address the stated objectives?

-Is the population clearly described and appropriate for the hypothesis being tested?

-Is the sample size sufficient to ensure adequate power to address the hypothesis being tested?

-Were correct statistical analysis used to support conclusions?

-Are there concerns about ethical or regulatory requirements being met?

Reviewer #1: The methods are clearly described.

Reviewer #2: (No Response)

**Results**

-Does the analysis presented match the analysis plan?

-Are the results clearly and completely presented?

-Are the figures (Tables, Images) of sufficient quality for clarity?

Reviewer #1: yes, the analysis matches the plan

The figures need to better label the x and y axes.

Reviewer #2: (No Response)

**Conclusions**

-Are the conclusions supported by the data presented?

-Are the limitations of analysis clearly described?

-Do the authors discuss how these data can be helpful to advance our understanding of the topic under study?

-Is public health relevance addressed?

Reviewer #1: The conclusions are supported by the data, however, the epidemiological patterns included in this analysis seem to assume these patterns cover the most common patterns. Unfortunately, cholera outbreaks run a large gamut of patterns from very brief outbreaks seen most commonly in Africa to the nearly continuous endemic pattern seen in Bangladesh. Thus, additional limitations of their model need to be described.

Reviewer #2: (No Response)

**Editorial and Data Presentation Modifications?**

Reviewer #1: Minor revisions

Reviewer #2: (No Response)

**Summary and General Comments**

Reviewer #1: Optimizing one-dose and two-dose cholera vaccine allocation in outbreak settings: A modeling study.

This is an elegant modeling study evaluating different dosing strategies for using oral cholera vaccine. They use the cholera epidemiological patterns from three geographic areas in an attempt to capture different patterns in the different regions. These three areas have cholera patterns that are quite unique. I am assuming that other reviewers who are more skilled in modeling methods will provide additional review of the actual modeling mathematics, so my review will mainly deal with comments on how this model may be used for decision making. 

It appears that the findings are quite consistent with earlier publications, especially “Azman AS, et al. The impact of a one-dose versus two-dose oral cholera vaccine regimen in outbreak settings: A modeling study. PLoS Medicine. 2015;12(8):e1001867.” This article which is cited previously concluded that a single dose strategy provides significant public heath benefit compared to two doses when the number of vaccine doses is limited in an outbreak situation, but it did not deal with more prolonged outbreaks or endemic situations which this new manuscript attempts to include. 

The authors correctly point out that rapid deployment of vaccine is key to a successful vaccine campaign during outbreaks. Unfortunately, when attempting to vaccinate quickly, the current procedures for obtaining vaccine from a central global stockpile, even though being designed to be rapid, in fact often result in vaccinations beginning delivered well after the peak of the outbreak and sometimes even after the outbreak is over (see “Bwire G, et al. Use of surveys to evaluate an integrated oral cholera vaccine campaign in response to a cholera outbreak in Hoima district, Uganda. BMJ Open 2020, 10:e038464”. It would seem that, although the authors do stress this aspect, the role of delays in vaccination might be stressed even more and a sensitivity analysis to account for delays would be valuable. 

 The authors should describe the three mechanisms for accessing OCV from the global stockpile. These include 1) emergency use to control outbreaks, 2) emergency use to prevent outbreaks during a humanitarian crisis, and 3) preventive use of vaccine for areas determined to be hot-spots. It is not clear which of these situations apply to their model and the manuscript would benefit from aligning their model with these three. 

For example, if one is using the vaccine for the emergency control of an outbreak, most outbreaks occur suddenly, and are of relatively short duration so nearly all the benefit occurs in the first year. Another outbreak may not occur for several years so determining a benefit over three years would show little impact relative to a one year impact and providing a second dose to this area will increase costs but provide little additional benefit. This would change of course, if this same area was also previously determined to be a hot-spot. (Outbreaks are more likely to occur in such hot-spots.) 

If the campaign is intending to vaccinate a “hot-spot” where outbreaks occur frequently, the second dose could more readily demonstrate additional benefit. Both of these situations are also then different from the endemic situation in Bangladesh where cholera occurs continuously at predicable rates. Obviously, a model cannot consider every possible scenario, but the discussion could provide additional context for these different situations. 

A specific queries and suggestions. 

For the age group <5, did the model assume that all children under age 5 would be vaccinated or only the age group >1 year. (The vaccine is not approved for children <1 year).

Reviewer #2: (No Response)

PLOS authors have the option to publish the peer review history of their article (what does this mean?). If published, this will include your full peer review and any attached files.

Reviewer #1: No

Reviewer #2: No

Figure Files:

Data Requirements:

Reproducibility:

References

---

## [Editor Report · Decision Letter 1]

25 Mar 2022

Dear Dr. Matrajt,

We are pleased to inform you that your manuscript 'Optimizing one-dose and two-dose cholera vaccine allocation in outbreak settings: A modeling study.' has been provisionally accepted for publication in PLOS Neglected Tropical Diseases.

Best regards,

James Michael Fleckenstein, M.D.

Associate Editor

Sharon Tennant

Deputy Editor

---

## [Editor Report · Acceptance letter]

14 Apr 2022

Dear Dr. Matrajt,

We are delighted to inform you that your manuscript, "Optimizing one-dose and two-dose cholera vaccine allocation in outbreak settings: A modeling study.," has been formally accepted for publication in PLOS Neglected Tropical Diseases.

Best regards,

Shaden Kamhawi

co-Editor-in-Chief

Paul Brindley

co-Editor-in-Chief
